# Asymptotic Optimality for Active Learning Processes

**Xueying Zhan**[1]             **Yaowei Wang**[2]             **Antoni B. Chan**[1]

[1] Department of Computer Science, City University of Hong Kong, Hong Kong SAR, China
[1] Department of Computing, The Hong Kong Polytechnic University, Hong Kong SAR, China

## Abstract

Active Learning (AL) aims to optimize basic learned model(s) iteratively by selecting and annotating unlabeled data samples that are deemed to best maximise the model performance with minimal required data. However, the learned model is easy to overfit due to the biased distribution (*sampling bias* and *dataset shift*) formed by non-uniform sampling used in AL. Considering AL as an iterative sequential optimization process, we first provide a perspective on AL in terms of statistical properties, i.e., asymptotic unbiasedness, consistency and asymptotic efficiency, with respect to basic estimators when the sample size (size of labeled set) becomes large, and in the limit as sample size tends to infinity. We then discuss how biases affect AL. Finally, we proposed a flexible AL framework that aims to mitigate the impact of bias in AL by minimizing generalization error and importance-weighted training loss simultaneously.

## 1 INTRODUCTION

The main goal of AL is to iteratively optimize a basic learning model with a finite set of data samples $\mathcal{D}_n = \{(\mathbf{x}_i, y_i)\}_{i=1}^n$, where each data sample $(\mathbf{x}_i, y_i)$ is sequentially selected from the unlabeled data pool and annotated. AL iterates between **data collection** and **model fitting** by repeatedly querying the labels of new data samples. Thus, selection of both the basic learning model and selection rule are of vital importance. From the perspective of data collection, AL has two branches: 1) pool-based AL, which selects new data sample(s) from a large unlabeled data pool for annotation, and 2) stream-based AL, which receives one data sample at a time and determines whether or not to label the instance [Cheng et al., 2013].

Many existing AL works focus on how to design acquisition functions based on **fixed heuristics** for data collection. For instance, uncertainty-based sampling strategies aim to select unlabeled data samples with the lowest confidence (the largest uncertainty) of being classified correctly by the basic model [Lewis and Catlett, 1994]. Most uncertainty-based methods belong to non-agnostic AL sampling strategies, that is, when making selections, the active learners rely more on the decision boundary estimated by the currently-trained basic model [Pereira-Santos et al., 2019]. In contrast, agnostic AL approaches make no assumption related to the decision boundary learned by the basic classifier, ignoring the information provided by the basic classifier (e.g., label information) and only utilizing the information directly from the unlabeled data pool [Pereira-Santos et al., 2019]. For instance, many representativeness-based methods, which select subsets that are most representative of the unlabeled data pool, are agnostic AL. Combined strategies [Shen et al., 2004, Ebert et al., 2012, Li and Guo, 2013, Ash et al., 2019] integrate the advantages of aforementioned sampling strategies, and are more adaptable to various data topologies [Munro, 2020].

However, the whole AL process is changing constantly with the labeled set and basic model updating in each stage, and thus it is not enough to just collect data "actively" and treat the model fitting stage in the same manner as passive learning. For passive learning, one key assumption is that the training set comprises *i.i.d.* (independent and identically distributed) samples from the unknown true data distribution $P(\mathbf{x}, y)$, $\mathcal{D}_n \overset{i.i.d.}{\sim} P$. If we select data samples sequentially by some fixed heuristics in AL (e.g., uncertainty-based strategies), the labeled training set is **not** drawn *i.i.d.* from $P$. That is, the labeled training set employed in AL is biased, due to the recycled use of past samples at each stage and the lack of independence between data samples [Fredlund et al., 2010, Portier and Delyon, 2018, Farquhar et al., 2021]. In this paper, we denote the bias resulting from the fixed heuristics during data collection as "*sampling bias*", which is inevitable during the whole AL processes.

Given a training set with *sampling bias*, an unbiased and

*Accepted for the 38th Conference on Uncertainty in Artificial Intelligence* (UAI 2022).

consistent estimator of a basic model in passive learning might no longer be unbiased (with respect to the original dataset), even asymptotically, in AL [Sugiyama and Nakajima, 2009, Farquhar et al., 2021]. For instance, *ordinary least squares* becomes biased due to the sampling bias problems [Sugiyama and Nakajima, 2009]. The combination of a biased data collection procedure and biased model estimator will result in a vicious cycle, where biased samples create biased models, then create even more biased samples. Dasgupta and Hsu [2008] provided a detailed explanation of this phenomenon: many AL heuristics start by utilizing a small initial labeled set to estimate a rough decision boundary, and then querying points that are increasingly closer to their current estimate of the boundary. During AL training, samples are queried based on increasingly confident assessments of their informativeness, e.g., the largest uncertainty, and the labeled set will diverge farther away from the true underlying data distribution. Moreover, if the basic learned model itself is biased, the rate of divergence will be accelerated.

In this paper, we explore the relationship between data collection and model fitting stages in AL, and discuss crucial factors for designing AL approaches that help reduce the negative effects of sampling bias. We then propose a flexible AL framework that can be applied on top of existing AL sampling schemes, through minimizing the combination of generalization error and re-weighted training loss in each stage. In our work, we utilize existing AL sampling schemes to generate sampling distribution. The sampling distribution is then used to compute the *importance weight* that represents the discrepancy between the underlying data distribution and the AL sampling distribution – that is – modeling the *dataset shift*. The importance weight is designed for minimizing the generalization error, and is used to re-weight the training loss for model learning in each stage. The re-weighted training loss is an asymptotically unbiased and consistent estimator of the true risk. Furthermore, this risk estimator could achieve asymptotic efficiency by optimizing its hyper-parameters.

## 2 RELATED WORK

One challenging and common problem in the research fields of learning from insufficient data (e.g., AL, few-shot learning, semi-supervised learning) is the overfitting of the learned models due to the biased distribution formed by limited training data. Bias appears in both stages of AL: *sampling bias*, which is attributed to the AL heuristics in data selection stage, and *dataset shift*, which is caused by the sampling bias and influences the model fitting stage.

*Sampling bias* is a bias in which samples are collected in a way that some samples of the intended population have a lower or higher sampling probability than others. If *sampling bias* is not accounted for, experiment results (e.g., performance of the basic model in AL) might be erroneously

attributed to the phenomenon or the model selection under study, rather than the method of sampling. On the one hand, sampling bias has a negative impact on AL as discussed in Section 1. Schütze et al. [2006] observed a "missed cluster effect" of AL, where some important clusters in the feature space are not represented in the AL sample set, and thus this sample set is not sufficient for estimating a basic model that is consistent with one learned from the true distribution. Furthermore, AL sampling will ignore these clusters in the data, and never query points from there, results in a local minimum. On the other hand, *sampling bias* sometimes can be helpful in AL. Mussmann and Liang [2018] proved that the uncertainty sampling updates are preconditioned SGD steps on the population $0/1$ loss, and move in descent directions for parameters that are not approximate stationary points. Chang et al. [2017] confirmed that the proper bias can be beneficial to generalization performance. They proposed "active bias" that emphasizes uncertain points and find that it increases the model performance, compared with using a fully labeled set.

*Dataset shift* (aka *covariate shift*, *dataset drifting*) refers to the discrepancy between the data distributions of the training and testing sets (or true underlying data distribution). It causes a principal problem during the model fitting step in AL, since some regions with large density in the unlabeled data pool may not be well represented by the labeled data. To reduce the impact of *dataset shift*, the labeled training set could be re-sampled with respect to an appropriate distribution, so as to minimize the statistical risk of the classifier built on the re-sampled data [Zadrozny, 2004, Richards et al., 2011]. Both *sampling bias* and *dataset shift* lead to error/bias in learning the optimal hypothesis. In this paper, we distinguish them to help explain why AL processes must be biased: the AL sampling strategy itself brings *sampling bias*, and this *sampling bias* creates the *dataset shift*.

To reduce bias problems in AL, Sener and Savarese [2017] provided a representativeness-based model that utilizes coreset approach (i.e., $k$-center) for pool-based AL, i.e. choosing set of points such that a model learned over the selected subset is competitive for the remaining data points. A rigorous bound between the average loss over the given subset and the remaining data points is derived by decomposing an upper bound of the AL loss. Inspired by Sener and Savarese [2017], we decompose the upper bound of the AL loss as a combination of training error and generalization error, but different from [Sener and Savarese, 2017], we focus on modeling the discrepancy between the true underlying data distribution and the AL sampling distribution.

Ganti and Gray [2012] and Imberg et al. [2020] proposed unbiased pool-based AL sampling schemes with the idea of "subroutine rejection-threshold" from [Beygelzimer et al., 2009] and the Horvitz-Thompson unbiased estimator [Horvitz and Thompson, 1952]. Ganti and Gray [2012] formally proved that *importance-weighted risk is an unbi-*

_ased estimator of the true risk_. Imberg et al. [2020] derived asymptotic Taylor expansions for the expected generalisation error and mean squared error of the predictions, and consequently presented sampling schemes that optimise the performance of AL approaches. Beygelzimer et al. [2009] proposed an importance-weighted AL sampling scheme based on the learner called a _subroutine rejection-threshold_, which efficiently corrects the sampling bias. However, due to the nature of the selected learner, their work is only applicable to stream-based AL. Farquhar et al. [2021] constructed an unbiased estimator of the empirical risk of the labeled set via the risk of unlabeled data pool with weighted loss. The aim is to remove the bias in AL, that is, they minimize the difference between the two risks, not the train-test data gap (_dataset shift_). In contrast, our work models the divergence between the underlying distribution of the whole data space and the AL sampling distribution, and we minimize a more general "train-test" gap – the gap between labeled data distribution and the true underlying data distribution.

# 3 METHODOLOGY

In this section, we firstly discuss crucial factors for designing AL methods that reduces the aforementioned bias problems. Then we propose our AL framework that can be applied on top of existing AL strategies.

## 3.1 AL LOSS

Assume we have an AL strategy $\mathcal{A}$ for a $K$-class classification task with feature space $\mathcal{X}$, label space $\mathcal{Y} \in \{1, ..., K\}$, classifier $f$ and a loss function $l(f(\mathbf{x}; \theta), y) : \mathcal{X} \times \mathcal{Y} \to \mathbb{R}$, parameterized over the hypothesis $\theta$. In general, passive learning aims to minimize the risk:

$$
\begin{aligned}
R(\theta) &= E_{(\mathbf{x},y) \sim P}[l(f(\mathbf{x}; \theta), y)] \\
&= \iint l(f(\mathbf{x}; \theta), y) P(\mathbf{x}, y) d\mathbf{x} dy.
\end{aligned} \tag{1}
$$

$P$ is unknown in most practical situations, but we can obtain sample data $\mathcal{D}_n$, and thus approximate it with an empirical distribution $P_\delta(\mathbf{x}, y) = \frac{1}{n} \sum_{i=1}^n \delta(\mathbf{x} = \mathbf{x}_i, y = y_i)$, where $\delta(\cdot)$ is a Dirac mass centered at $(\mathbf{x}_i, y_i)$ [Zhang et al., 2017]. Then, the **empirical risk** is formulated as

$$
\begin{aligned}
R^{\mathrm{emp}}(\theta) &= \iint l(f(\mathbf{x}; \theta), y) P_\delta(\mathbf{x}, y) d\mathbf{x} dy \\
&= \frac{1}{n} \sum_{i=1}^n l(f(\mathbf{x}_i; \theta), y_i).
\end{aligned} \tag{2}
$$

Generally, in statistical machine learning, there is an assumption that the empirical risk will converge to the true risk as the number of samples increases, i.e., the empirical risk is an asymptotically unbiased estimate of the true risk, as follows:

**Lemma 1** _The empirical risk_ $R^{\mathrm{emp}}(\theta)$ _is an **asymptotically unbiased** estimate of_ $R(\theta)$:

$$
\lim_{n \to \infty} E[R^{\mathrm{emp}}(\theta) - R(\theta)] = 0. \tag{3}
$$

In AL, consider a large unlabeled data pool $\mathcal{D}_u$ that is sampled _i.i.d._ from $P$, where the label of each data is unobserved, and an initial labeled data pool $\mathcal{D}_0 = \{(\mathbf{x}_i, y_i)\}_{i=1}^{n_0}$ that is also sampled _i.i.d._ from $P$. Strategy $\mathcal{A}$ sequentially selects an unlabeled subset from $\mathcal{D}_u$ and queries their labels from an oracle for building the training set. At stage $t$, we collect and label new data samples $\mathcal{D}_t^{new}$ from $\mathcal{D}_u$, where $|\mathcal{D}_t^{new}| = n_t - n_{t-1} \leq B$ and $B$ is the batch size. We then update the training set, obtaining $\mathcal{D}_t = \{(\mathbf{x}_i, y_i)\}_{i=1}^{n_t}$.

In most existing AL works, following multiple stages of a myopic approach, where each stage is solved independently, the goal of AL for a single stage is written as:

1. Select and label new samples: $\mathcal{D}_t^{\mathrm{new}} = \arg\min_{\mathbf{x} \in \mathcal{D}_u}^B a(\mathbf{x}; \mathcal{A}|\mathcal{D}_{t-1})$, where $a(\mathbf{x}; \mathcal{A})$ is the acquisition function of AL strategy $\mathcal{A}$ (and parameters within) [Gal et al., 2017], and update $\mathcal{D}_t = \mathcal{D}_t^{\mathrm{new}} \cup \mathcal{D}_{t-1}$.
2. Calculate the empirical risk at stage $t$, denoted as $R_t^{\mathrm{emp}}(\theta)$ [Sener and Savarese, 2017]:

$$
R_t^{\mathrm{emp}}(\theta) = \frac{1}{n_t} \sum_{i=1}^{n_t} l(f(\mathbf{x}_i; \theta), y_i), \tag{4}
$$

and minimize it to obtain the estimated optimal hypothesis at stage $t$: $\hat{\theta}_t = \arg\min_\theta R_t^{\mathrm{emp}}(\theta)$.

However, as mentioned in Section 2, an AL sampling heuristic will deliberately select a subset of unlabeled samples for labeling. Thus the selected data set will be distributed differently from the true distribution $P(\mathbf{x}, y)$. We denote the data distribution _induced_ by the AL algorithm as an _instrumental_ distribution $Q(\mathbf{x}, y)$. Therefore, the risk estimation during AL is actually biased since the labeled set is sampled from $Q$ instead of $P$. In this paper, we assume that _AL will not query non-existent or out-of-distribution (OOD) data samples, and oracles/experts will not produce wrong/noisy labels, that is, $P(\mathbf{x}, y) > 0$ and $Q(\mathbf{x}, y) > 0$._

Importance Weight Empirical Risk Minimization (IWERM) is widely adopted to remove bias in AL [Shimodaira, 2000, Sugiyama et al., 2007, Cortes et al., 2010, Vogel et al., 2020]. It is originally designed to solve _dataset shift_ [Shimodaira, 2000, Sugiyama et al., 2007, Sugiyama and Nakajima, 2009, Sawade et al., 2010, Vogel et al., 2020]. We denote _importance weight_ as $\beta(\mathbf{x}, y) = \frac{P(\mathbf{x}, y)}{Q(\mathbf{x}, y)}$. After reweighing by $\beta$, the **weighted empirical risk at stage $t$ under the $Q$ distribution** (denote as $R_t^w(\theta)$) is re-estimated:

$$
R_t^w(\theta) = \frac{1}{n_t} \sum_{i=1}^{n_t} \beta(\mathbf{x}_i, y_i) l(f(\mathbf{x}_i; \theta), y_i). \tag{5}
$$

Reweighing the empirical risk under the $Q$ distribution

forms an *unbiased* estimator of the true risk:

$$E_{(\mathbf{X},\mathbf{y})\sim Q}\left[\frac{1}{n_t}\sum_{i=1}^{n_t}\beta(\mathbf{x}_i,y_i)l(f(\mathbf{x}_i;\theta),y_i)\right]$$

$$= \frac{1}{n_t}\sum_{i=1}^{n_t}E_{(\mathbf{X},\mathbf{y})\sim Q}\left[\beta(\mathbf{x}_i,y_i)l(f(\mathbf{x}_i;\theta),y_i)\right]$$

$$= \frac{1}{n_t}\sum_{i=1}^{n_t}E_{(\mathbf{x}_i,y_i)\sim Q}\left[\beta(\mathbf{x}_i,y_i)l(f(\mathbf{x}_i;\theta),y_i)\right]$$

$$= \frac{1}{n_t}\sum_{i=1}^{n_t}\iint Q(\mathbf{x}_i,y_i)\frac{P(\mathbf{x}_i,y_i)}{Q(\mathbf{x}_i,y_i)}l(f(\mathbf{x}_i;\theta),y_i)d\mathbf{x}_i dy$$

$$= \frac{1}{n_t}\sum_{i=1}^{n_t}\iint P(\mathbf{x}_i,y_i)l(f(\mathbf{x}_i;\theta),y_i)d\mathbf{x}_i dy$$

$$= \frac{1}{n_t}\sum_{i=1}^{n_t}R(\theta) = R(\theta),$$

where $\mathbf{X} = \{\mathbf{x}_1,\cdots,\mathbf{x}_{n_t}\}$ and $\mathbf{y} = \{y_1,\cdots,y_{n_t}\}$.

## 3.2  UPPER BOUND OF AL LOSS

From (4) and (5), we observe that AL will perform well if we have: (i) larger labeled set, i.e., more budget; (ii) consistent and unbiased estimator of the empirical risk; (iii) considering (5), correctly modeling the discrepancy between $P$ and $Q$, which also accelerates the convergence rate of AL. Based on these considerations, to design a flexible AL approach, inspired by Sener and Savarese [2017], we consider an upper bound of the risk $R(\theta)$ using the triangle inequality:

$$R(\theta) \le \underbrace{\left|R(\theta) - R_t^w(\theta)\right|}_{\text{1st term}} + \underbrace{\left|R_t^w(\theta)\right|}_{\text{2nd term}}, \qquad (6)$$

The 1st term is the generalization error of the AL training process, while the 2nd term is the training loss, as in (5). In practice, the size of labeled set is finite and hence the loss function is bounded. Hoeffding's Inequality can quantify how these factors (i.e., $n_t$, $\beta$) affect the convergence of the 1st term, yielding the following theorem (see proof in supplementary materials).

**Theorem 1** (*Hoeffding Inequality with IWERM*) *Let* $\{(\mathbf{x}_i,y_i)\}_{i=1}^{n_t}$ *be* $n_t$ *instances that are sampled from the instrumental distribution* $Q(\mathbf{x},y)$. *Denote r.v.* $\mathbf{S} = R(\theta) - R_t^w(\theta)$ *that takes over* $\theta$, *and let* $b = \sup \mathbf{S}$, $a = \inf \mathbf{S}$, $E[\mathbf{S}] = \eta$. $\forall \epsilon > 0$, *we have*

$$\mathbb{P}\left(\left|R(\theta) - R_t^w(\theta)\right| \ge \epsilon\right) \le 2e^{\frac{-2n_t(\epsilon-\eta)^2}{(b-a)^2}}. \qquad (7)$$

When $n_t \to \infty$, $2\exp(\frac{-2n_t(\epsilon-\eta)^2}{(b-a)^2}) \to 0$, and thus the risk estimator $R_t^w(\theta)$ is also *consistent*. With ideal $\beta$, then $\eta = 0$ since $R_t^w(\theta)$ is an unbiased estimate of $R(\theta)$. Additionally, for the vanilla risk case $R_t^{\text{emp}}(\theta)$ (when $\beta = 1$ in $R_t^w(\theta)$), which is the case with most AL methods, under an AL scenario with finite labeled set, $P$ is not generally equal to $Q$, and thus $\eta \ne 0$ during the AL process. Thus, $R_t^{\text{emp}}(\theta)$

is **not** an unbiased and consistent estimate of the risk $R(\theta)$ [Shimodaira, 2000]. This shows the superiority of IWERM.

For the 2nd term in (6), the empirical risk of the selected samples are weighted appropriately to compensate for the discrepancy between the instrumental and true distributions, which leads to a consistent and asymptotically unbiased estimate of the risk [Sawade et al., 2010]. Previous IWERM works [Sugiyama et al., 2007, Sugiyama and Nakajima, 2009, Sawade et al., 2010, Vogel et al., 2020] assume that the source data distribution for training is different with the target data distribution for testing, but the difference only comes from the input distributions $P(\mathbf{x})$ and $Q(\mathbf{x})$, while the posterior distributions $P(y|\mathbf{x})$ and $Q(y|\mathbf{x})$ are assumed to be identical. In our work, we **relax this assumption** and consider that the full joint distributions $P(\mathbf{x},y)$ and $Q(\mathbf{x},y)$ are different.

## 3.3  IMPORTANCE WEIGHT ESTIMATION

The estimation of IWERM is both unbiased and consistent using ideal importance weight $\beta(\mathbf{x},y)$. However, the ideal $\beta$ is not achievable in practice, and thus we approximate $\beta(\mathbf{x},y)$ by $\beta_t(\mathbf{x},y)$ in every stage $t$. The estimator based on $\beta_t(\mathbf{x},y)$ is still consistent and asymptotically unbiased under the following conditions:

$$\beta_t(\mathbf{x},y) = \frac{P_t(\mathbf{x},y)}{Q_t(\mathbf{x},y)} \to \frac{P(\mathbf{x},y)}{Q(\mathbf{x},y)} = \beta(\mathbf{x},y), \text{ as } n_t \to \infty. \qquad (8)$$

(8) holds by properly selecting the formulation of $P_t$ and $Q_t$, which will be introduced in the next sections. Specifically, if $\lim_{n_t\to\infty}\beta_t(\mathbf{x},y) = 1$ holds, then $P(\mathbf{x},y) = Q(\mathbf{x},y)$ as sample size tends to infinity. Thus, the estimator $R_t^w(\theta)$ is asymptotically unbiased and consistent for AL sampling strategies that converge to "non-informativeness", which is defined as follows:

**Definition 1** (*Non-informativeness*) *An acquisition function* $a(\cdot;\theta)$ *is "non-informativeness" if the output is a constant (denoted as* $c_a$) *for arbitrary input* $\mathbf{x}$:

$$\lim_{n_t\to\infty}a(\mathbf{x};\theta) = c_a, \forall \mathbf{x} \in \mathcal{X}. \qquad (9)$$

We will explain the reason in Section 3.3.3.

### 3.3.1  $P$ Distribution

Since $P(\mathbf{x},y)$ is unknown in practice, in Bayesian inference, a family of probability distributions $P_t(\mathbf{x},y|\phi)$ is specified to approximate $P(\mathbf{x},y)$ [Box and Tiao, 2011, Tran, 2017], where $\phi$ is not known in advance and needs to be estimated from observed data samples. At stage $t$, given labeled set $\mathcal{D}_t = \{(\mathbf{x}_i,y_i)\}_{i=1}^{n_t}$ and a prior distribution $p(\phi)$, the posterior distribution of the parameter $\phi$ is estimated as

$$P_t(\phi|\mathcal{D}_t) = \frac{p(\phi)\prod_{i=1}^{n_t}P_t(\mathbf{x}_i,y_i|\phi)}{\int p(\phi)\prod_{i=1}^{n_t}P_t(\mathbf{x}_i,y_i|\phi)d\phi}. \qquad (10)$$

The predictive distribution is

$$P_t(\mathbf{x}, y|\mathcal{D}_t) = \int P_t(\mathbf{x}, y|\phi) P_t(\phi|\mathcal{D}_t) d\phi. \quad (11)$$

### 3.3.2 $Q$ Distribution

Inspired by [Fredlund et al., 2010], rather than selecting a particular datum to query, we model the query as a draw of a sample from the distribution $Q$. We define the $Q$ distribution at stage $t$ as follows:

$$Q_t(\mathbf{x}, y) = Q_t(\mathbf{x}, y; \theta) = \frac{q_t(\mathbf{x};\theta)P_t(\mathbf{x},y)}{\iint q_t(\mathbf{x};\theta)P_t(\mathbf{x},y)d\mathbf{x}dy}. \quad (12)$$

$q_t(\mathbf{x}; \theta)$ is an AL querying density function, where $q_t(\mathbf{x};\theta) > 0, \forall \mathbf{x} \in \mathcal{X}$ and $\int q_t(\mathbf{x};\theta)d\mathbf{x} = 1$. Note that $Q_t$ is relative to the underlying distribution, i.e., specifies the relative over-sampling or under-sampling w.r.t. $P$. Choosing $q_t$ to be constant is equivalent to selecting instances at random from the data pool (uniform sampling). In contrast, choosing $q_t$ to be narrow will focus AL on a particular region, and in the limit, setting $q_t$ to a delta function will select a particular sample without reference to the underlying distribution $P$ [Fredlund et al., 2010].

In pool-based AL, we select instances based on maximizing the acquisition function $a(\mathbf{x}; \mathcal{A})$: $\mathbf{x}^* = \arg\max_{\mathbf{x} \in \mathcal{D}_u} a(\mathbf{x}; \mathcal{A})$, and thus the acquisition function should be converted into a querying density. For example, entropy-based uncertainty methods will select data samples with the largest entropy across all classes, and the corresponding acquisition function is: $a(\mathbf{x}) = \sum_{k=1}^{K} \bar{p}(y = k|\mathbf{x};\theta) \log \bar{p}(y = k|\mathbf{x};\theta)$, and $\bar{p}$ is the predicted class probability of given $\mathbf{x}$.

In our work, we convert the acquisition function to querying density function $q_t$ by applying the *softmax* function

$$q_t(\mathbf{x}_i; \theta) = \frac{\exp(\alpha_t a(\mathbf{x}_i; \mathcal{A}_t))}{\sum_j \exp(\alpha_t a(\mathbf{x}_j; \mathcal{A}_t))}, \quad (13)$$

where $\alpha_t$ is temperature hyperparameter ($\alpha_t > 0$), and $\mathcal{A}_t$ is the AL strategy at stage $t$. Note that the softmax does not change the ranking of the unlabeled samples for AL sampling. We select the temperature hyperparameter $\alpha_t$ to preserve the asymptotic efficiency of the whole AL processes (see Proposition 2 in Section 3.3.4).

Finally, we approximate $Q(\mathbf{x}, y)$ by $Q_t(\mathbf{x}, y; \theta, \phi)$, w.r.t. $q_t$ and $P_t(\mathbf{x}, y|\phi)$, as follows:

$$Q_t(\mathbf{x}, y; \theta, \phi) = \frac{q_t(\mathbf{x};\theta)P_t(\mathbf{x},y|\phi)}{\iint q_t(\mathbf{x};\theta)P_t(\mathbf{x},y|\phi)d\mathbf{x}dy}. \quad (14)$$

### 3.3.3 Importance Weight $\beta$

Next we represent the approximation of $\beta(\mathbf{x}, y)$ as $\beta_t(\mathbf{x}, y; \theta, \phi)$ at stage $t$ as

$$\beta_t(\mathbf{x}, y; \theta, \phi) = \frac{P_t(\mathbf{x},y|\phi)}{Q_t(\mathbf{x},y;\theta,\phi)} = \frac{\iint q_t(\mathbf{x};\theta)P_t(\mathbf{x},y|\phi)d\mathbf{x}dy}{q_t(\mathbf{x};\theta)}. \quad (15)$$

We analyse the representation of $\beta_t$ when the sample size tends to infinity. Supposing that $R_t^w(\theta)$ is an unbiased estimator of $R(\theta)$, which is based on a sufficiently strong classifier (e.g., using CNN as a basic classifier). If an infinite number of samples are observed, the basic classifier will have a very certain prediction given $\mathbf{x}_i$. Thus, AL sampling strategies like entropy-based uncertainty sampling will converge to "non-informativeness" as the sample size tends to infinity, since all predictions are certain. Note that the numerator in (15) is a constant w.r.t. $(\mathbf{x}, y)$. However, the numerator is still required for numerical stability, as setting it to 1 will yield large $\beta_t$ values that make the loss numerically unstable. Consider Assumption 1 below, in which case the acquisition function will output a constant for any $\mathbf{x}$ as the sample size increases.

**Assumption 1** *The existing AL strategy $\mathcal{A}$ adopted in AL querying density function converge to "non-informative" as the sample size increases.*

Under Assumption 1, $\lim_{n_t \to \infty} q(\mathbf{x}_i; \theta) = \frac{\exp(\alpha_t c_a)}{|\mathcal{D}_u|\exp(\alpha_t c_a)} = \frac{1}{|\mathcal{D}_u|}$ in (13), where $|\mathcal{D}_u|$ is the size of unlabeled pool[1]. Then, (15) becomes

$$\lim_{n_t \to \infty} \beta_t(\mathbf{x}, y; \theta, \phi) = \frac{\iint (1/|\mathcal{D}_u|)P_t(\mathbf{x},y|\phi)d\mathbf{x}dy}{1/|\mathcal{D}_u|} = 1, \quad (16)$$

where $\iint P_t(\mathbf{x}, y|\phi)d\mathbf{x}dy = 1$ since $P_t(\mathbf{x}, y|\phi)$ is a probability density function. Thus, the whole process is asymptotically unbiased and consistent.

Note that (8) converges point-by-point based on our assumption of "non-informativeness" and our designed $P_t$ and $Q_t$. The reasons are as follows. First, in the ideal case, as sample size tends to infinity, enough data is observed and thus the underlying data distribution $P$ is known. Thus, the optimal sampling distribution should be the data distribution itself, i.e., $P_i = Q_i$, and thus $\beta_i = P_i/Q_i = 1$. Second, based on (13) and our "non-informativeness" requirement, $Q_t$ would also be the same as $P_t$ as sample size tends to infinity, since $q_t$ converges to a constant, and thus $\beta_i = P_i/Q_i = 1$. Regarding the convergence of $\beta_t$ in (15), we can regard the numerator $\iint q_t(\mathbf{x};\theta)P_t(\mathbf{x},y|\phi)d\mathbf{x}dy$ as a normalization constant. In the remaining part $\frac{1}{q_t(\mathbf{x};\theta)}$, $q_t$ is the softmax function (see (13)), which tends to 0 if and only if $q_t \to +\infty$. However, this condition will never be satisfied according to Section 4.2 in [Guo et al., 2017]. Thus, $\beta_t$ will converge to a finite value.

We further explain why some AL methods can converge to "non-informativeness" based on the assumptions in AL sampling processes from two aspects, using entropy-based uncertainty sampling as example. We assume that AL would not query non-existing or out-of-distribution (OOD) data

---

[1]See more discussions and examples of "non-informativeness" AL sampling strategies in supplementary materials.

samples and would not query wrong/noisy labels from oracles/experts, that is, $P(\mathbf{x}, y) > 0$ and $Q(\mathbf{x}, y) > 0$. Additionally, we could also obtain another vital information from these assumptions: $P(y = y_{\text{true}}|\mathbf{x}_i) = 1$ for all labeled samples. Firstly, after querying enough samples, any $\mathbf{x}_i$ actually appears in the labeled trained set, and thus we know the hard label and are very certain about it, *i.e.*, $P(y = y_{\text{true}}|\mathbf{x}_i) = 1$, thus the confidence is 1. Secondly, [de Cossio and de Cossio Diaz, 2015] shows that the practice of using sample average as surrogates of probability expectations is reliable provided sample size is large. Equation (1) in [de Cossio and de Cossio Diaz, 2015] shows that the entropy of model parameters will converge to a certain value as sample size increases. That is, after observing enough data, any given $\mathbf{x}_i$ will not change the basic model, and thus any $\mathbf{x}_i$ is meaningfulness to improve the basic model, which is consistent with our proposed "non-informativeness" assumption. More discussions are in Appendix.

### 3.3.4   Parameter Estimation of $\alpha$

When calculating the $Q_t$ distribution, the temperature scaling parameter $\alpha_t$ in (13) needs to be estimated. We propose a method for estimating $\alpha_t$ by considering the asymptotic efficiency of $R_t^w(\theta)$, i.e., the asymptotic variance of the estimator (see proof of Proposition 1 in appendix).

**Proposition 1** *(Asymptotic Variance of Estimators) Let $R_t^w(\theta)$ be defined in (5) and $R(\theta)$ be defined in (1), by employing the "Delta Method", we have*

$$\sqrt{n_t}(R_t^w(\theta) - R(\theta)) \stackrel{n_t \to \infty}{\longrightarrow} \mathcal{N}(0, \sigma_Q^2), \qquad (17)$$

*with $\sigma_Q^2 = \iint \beta(\mathbf{x}, y)[l(f(\mathbf{x}; \theta), y) - R(\theta)]^2 P(\mathbf{x}, y)d\mathbf{x}dy$.*

We next consider selecting the parameters $\alpha_t$ so as to minimize the variance of the estimator (see proof of Proposition 2 in Appendix).

**Proposition 2** *(Optimal Sampling Distribution) The optimal instrumental sampling distribution that minimizes $\sigma_Q^2$ is*

$$Q_t^{opt}(\mathbf{x}, y) \propto \left| l(f(\mathbf{x}; \theta), y) - R(\theta) \right| P(\mathbf{x}, y). \qquad (18)$$

In practical use, we employ $P_t$ to approximate $P$, and thus, $Q_t^{opt} \approx \left| l(f(\mathbf{x}; \theta), y) - R(\theta) \right| P_t(\mathbf{x}, y)$. For each sample $(\mathbf{x}_i, y_i)$, define shorthand $Q^i(\alpha) = Q_t(\mathbf{x}_i, y_i; \theta, \phi, \alpha_t)$, $P^i = P_t(\mathbf{x}_i, y_i, |\phi)$, $l_i = l(f(\mathbf{x}_i; \theta), y_i)$, and $R = R(\theta)$. Based on (18), to obtain the optimal sampling distribution, we set $\frac{Q^i(\alpha_t)}{|l_i - R|P^i} = c_o$ for some constants $c_o$. Equivalently taking the logarithm, $\log Q^i(\alpha_t) - \log |l_i - R|P^i = \log c_o$.

---

**Algorithm 1** The proposed AL Framework.

---

**Require:** Initial labeled set $\mathcal{D}_0$, unlabeled data pool $\mathcal{D}_u$, prior information of $p(\phi)$, AL method $\mathcal{A}$, initial importance weight $\beta_0 = \{1, 1, ...\}$, batch size $B$, oracle $\mathcal{O}$.

1: Stage 0: Estimate initial model $\hat{\theta}_0 = \min_\theta R_0(\theta)$ with $\mathcal{D}_0$ and $\beta_0$. Estimate $\hat{\phi}_0$ from $\mathcal{D}_0$. Estimate $q_0(\mathbf{x}; \hat{\theta}_0)$. Calculate $Q_0(\mathbf{x}, y; \hat{\theta}_0, \hat{\phi}_0)$.
2: **for** stage $t$ in $1, ..., T$ **do**
3:    *Update labeled set*: Obtain $B$ data samples from $\mathcal{D}_u$ with $Q_{t-1}$, and query labels from $\mathcal{O}$. Add the new samples $\mathcal{D}_t^{\text{new}}$ to $\mathcal{D}_{t-1}$ to obtain $\mathcal{D}_t$.
4:    *Update unlabeled set*: update $\mathcal{D}_u$ by removing $\mathcal{D}_t^{\text{new}}$.

5:    *Estimate $P_t$*: Estimate $\hat{\phi}_t$ with $\mathcal{D}_t$ by (10). Calculate $P_t(\mathbf{x}, y)$ by (11).
6:    *Estimate $Q_t$*: Update $\alpha_t$ by (19). Calculate $q_t(\mathbf{x}; \hat{\theta}_{t-1})$ by (13) and $Q_t(\mathbf{x}, y; \hat{\theta}_{t-1}, \hat{\phi}_t)$ by (14).

7:    *Importance weight:* Calculate $\beta_t(\mathbf{x}, y)$ by (15).
8:    *Re-train basic model(s):* $\hat{\theta}_t = \min_\theta R_t^w(\theta)$ from (5).
9: **end for**

---

Thus, $(\alpha, c_o)$ can be estimated by minimizing the squared error of the log-constant term, summed over all labeled samples, at stage $t$, we have

$$\alpha_t^*, c_o^* = \arg\min_{\alpha_t, c_o} \sum_i (\log Q^i(\alpha_t) - \log |l_i - R|P^i - \log c_o)^2.$$
$$(19)$$

Note that $R(\theta)$ is generally expected to be a very small value close to zero for a well-trained model [Sener and Savarese, 2017]. In our experiments, we set $R = 10^{-3}$. There is no closed-form solution, and instead we use a numerical optimization toolbox[2] to solve for $\alpha_t, c_o$ in each stage $t$.

### 3.4   PROPOSED AL FRAMEWORK

In summary, we propose a flexible AL framework on top of existing AL strategies based on IWERM. The proposed framework gives an asymptotically unbiased and consistent estimate of the true risk if Assumption 1 holds, which can be satisfied by proper selection of the AL strategy and the basic model. Additionally, the hyperparameter $\alpha_t$ is selected to minimize the variance of the risk estimator, and thus our framework is also asymptotically efficient. The whole framework is described in Algorithm 1.

## 4   EXPERIMENT

To validate the effectiveness of our proposed AL framework, we compare the performance between existing AL strategies

---

[2]E.g., minimize function in Scipy library.

(as baseline methods) and incorporated with our unbiased AL framework (as the basic AL acquisition functions). We also compare our model with other de-biased/less biased AL sampling schemes.

## 4.1 EXPERIMENTAL SETTINGS

### 4.1.1 Datasets

We consider 8 datasets for classical ML tasks: 4 real-life UCI datasets [Dua and Graff, 2017], including *Clean1*, *Splice*, *Tic-tac-toe*, and *Vehicle*; 4 synthetic datasets, including *EX8a* [Ng, 2008], *Gaussian Cloud Unbalance* [Konyushkova et al., 2017], *R15* and *D31* [Veenman et al., 2002]. The datasets can be categorized into: synthetic data (*EX8a*, *GCloudub*, *R15* and *D31*); real-life data (*Clean1*, *Splice*, *Tic-tac-toe* and *Vehicle*); binary-class classification tasks (*EX8a*, *GCloudub*, *Clean1*, *Splice* and *Tic-tac-toe*); multi-class classification tasks (*R15*, *D31*, and *Vehicle*); imbalanced data cases (*GCloudub* and *Tic-tac-toe*).

### 4.1.2 Baselines

We compare our model with 4 typical AL strategies [Settles, 2009], including entropy-based Uncertainty Sampling (**US**) [Lewis and Catlett, 1994], Query-by-Committee (**QBC**) [Seung et al., 1992], Expected Error Reduction (**EER**) [Roy and McCallum, 2001] and Batch-mode Discriminative and Representative AL (**BMDR**) [Wang and Ye, 2015]. **US** finds unlabeled data samples with largest entropy of predicted probabilities. **QBC** minimizes the version space (set of hypotheses that are consistent with labeled set). **EER** selects data points with minimal expected future risk. **BMDR** queries a batch of informative and representative examples by minimizing the empirical risk bound of AL. We also utilize these four AL strategies as basic AL methods in our framework by using (13). We change the output of these AL methods (the data samples to query, ranked by the corresponding acquisition function) to the querying density (normalizing the actual output of the acquisition function for unlabeled data samples).

We also compare our proposed method with 2 unbiased AL sampling methods, which are based on importance sampling/weighting techniques: Unbiased Pool-based AL (**UPAL**) [Ganti and Gray, 2012] and Sampling-Weighted AL (**SWAL**) [Imberg et al., 2020], which has 3 variants: **SWAL-cora** (Corollary 1 (a) in Imberg et al. [2020]), **SWAL-corb** (Corollary 1 (b) in Imberg et al. [2020]) and **SWAL-prop** (Proposition 1 in Imberg et al. [2020]). The implementations of **US**, **QBC**, **EER** and **BMDR** are from ALiPy [Tang et al., 2019]. **UPAL** and **SWAL** are re-implemented with reference to the released code[3].

---

[3] https://github.com/imbhe/OSiUAL

### 4.1.3 Implementation Details

We repeated each experiment 10 times with randomly split training and testing sets, and reported the average testing performance. We employed the same basic classifier for the AL baselines and our methods under each dataset. To evaluate average performance, we compute area under the performance-budget curve (AUBC) [Zhan et al., 2021], by evaluating the AL method for different fixed budgets (e.g., Accuracy vs. Budget in Figure 1). The area under the curve is calculated by trapezoid method, with higher values reflecting better performance of AL under varying budgets. More details about experimental design are in the supplemental materials, including how $P$ is modeled (Section A3.3 and Section A4.3 in supplemental materials) the description of datasets, baselines and more implementation details (Section A4.1-4.3 in supplemental materials).

## 4.2 EXPERIMENTAL RESULTS

Figure 1 presents the accuracy-budget curves with batch size 10, with the AUBC values reported in the legend. Note that in these experiments, the size of $\mathcal{D}_u$ is set as the upper bound of the AL budget. That is, the basic AL models, **US**, **QBC**, **EER** and **BMDR** converge to the same accuracy at the end of the AL process, since their basic classifier will be trained on the whole training set with uniform importance weight (the vanilla risk case).

We next analyze the experimental results w.r.t. different dataset properties. More experimental results with different batch size settings ($B \in \{1, 5, 20\}$) and for various evaluation metrics (AUBC-AUC and AUBC-$F_1$) are presented in the supplementary materials (see Section A4.4).

### 4.2.1 Comparisons with Basic AL Methods

The purpose of this experiment is to observe if our proposed approach can enhance existing AL models. Comparing with basic AL sampling strategies, our approach significantly improves the performance of the basic AL methods, by achieving faster convergence. Especially, on *R15* (see Fig. 1c), our approaches converge after querying 10 samples, while the basic AL methods converge after 130 samples. On *D31* (Fig. 1d), the improvements are more significant – our approaches converge after 40 samples, while the basic AL methods gradually converge after 600 to 900 samples. Both *R15* and *D31* have clear data/cluster distributions, but the tasks are more difficult because there are more classes (*R15* has 15 classes and *D31* has 31 classes). The basic AL methods more easily fall into local optimum and make wrong judgments of the decision boundary, while our methods avoids this problem by correctly modeling the discrepancy between the underlying data distribution and the current sampling distribution, and thus achieves better performance.

Table 1: Comparison of our model (**BMDR**-based) against unbiased AL baselines. The table shows the mean and standard deviation AUBC (acc) values, and the highest AUBC (acc) values are in **bold**. A paired t-test was conducted between our method and the others, and *, **, *** indicate statistical significant differences at $p < 0.05$, $p < 0.01$, and $p < 0.001$, respectively. This experiment uses 10 trials and $B = 10$.

| Dataset | SWAL-cora | SWAL-corb | SWAL-prop | UPAL | BMDR-ours |
|---------|-----------|-----------|-----------|------|-----------|
| *EX8a* | $0.825 \pm 0.014$** | $0.819 \pm 0.018$** | $0.832 \pm 0.008$** | $0.841 \pm 0.016$ | $\mathbf{0.849 \pm 0.014}$ |
| *GCloudub* | $0.945 \pm 0.006$* | $0.943 \pm 0.010$* | $0.946 \pm 0.009$ | $0.946 \pm 0.008$* | $\mathbf{0.949 \pm 0.007}$ |
| *R15* | $0.749 \pm 0.053$*** | $0.733 \pm 0.036$*** | $0.889 \pm 0.023$*** | $0.881 \pm 0.035$*** | $\mathbf{0.979 \pm 0.006}$ |
| *D31* | $0.908 \pm 0.013$*** | $0.908 \pm 0.012$*** | $0.940 \pm 0.005$*** | $0.933 \pm 0.007$*** | $\mathbf{0.968 \pm 0.004}$ |
| *Clean1* | $0.795 \pm 0.017$* | $0.803 \pm 0.019$** | $0.785 \pm 0.034$* | $0.799 \pm 0.025$* | $\mathbf{0.815 \pm 0.022}$ |
| *Splice* | $0.785 \pm 0.014$* | $0.786 \pm 0.013$* | $0.788 \pm 0.014$ | $0.784 \pm 0.013$* | $\mathbf{0.795 \pm 0.013}$ |
| *Tic-tac-toe* | $0.763 \pm 0.016$ | $0.765 \pm 0.017$ | $0.762 \pm 0.016$ | $0.765 \pm 0.018$ | $\mathbf{0.768 \pm 0.021}$ |
| *Vehicle* | $0.679 \pm 0.012$** | $0.681 \pm 0.010$* | $0.671 \pm 0.017$* | $0.686 \pm 0.015$ | $\mathbf{0.692 \pm 0.010}$ |

On *GCloudub* (Fig. 1b), our approaches converge after 80 samples, while for the basic AL methods, **US**, **EER**, and **BMDR** converge after 100, 260, and 200 samples, respectively. Besides faster convergence, our method also provides more stable performance due to de-biasing the *dataset shift*. These basic AL sampling strategies do not always perform well on various data types, e.g., **US** and **EER** even show a performance drop on *Splice* (Fig. 1d). This is caused by *sampling bias*, the incorrect judgment of decision boundaries, as mentioned in Section 1, while our method reduces the effect of the *sampling bias* by correctly modeling the discrepancy between the sampling distribution and the underlying data distribution – the *dataset shift*. Our method improves the AUBC(acc) performances of **US** from 0.711 to 0.793 and **EER** from 0.716 to 0.794.

On datasets with class imbalance, the improvement from our approach is more substantial, e.g., on *GCloudub* with imbalance ratio (IR) 2.0, and on *Tic-tac-toe* with IR 6.8. For instance, based on AUBC (acc), we improve **BMDR** from 0.923 to 0.949 on *GCloudub*, improve **US** from 0.714 to 0.772 and improve **EER** from 0.716 to 0.766 on *Tic-tac-toe*. The better performance on imbalanced data is likely because of the importance-weighting, which reduces bias caused by under-sampling the larger class.

### 4.2.2 Comparison with Unbiased AL Methods

Comparing our approach with other unbiased AL sampling strategies (3 variants of **SWAL**, and **UPAL**), we observe that all these methods reduce the *sampling bias* problems during the AL process. Our method achieve the best performance on all of the 8 datasets, as shown in Fig. 2. We further examine the differences in performance between our method (we choose **BMDR**) and unbiased AL baselines (**SWAL** and **UPAL**) by using a paired t-test on 10 repeated trials on each dataset. The test results are shown in Table 1. Our method outperforms the baselines at a statistically significant level ($p < 0.05$) on 24 out of 32 (75%) of the experiments, while

performing similarly ($p > 0.05$) on 8 out of 32 (25%). The t-test results indicate that, compared with the baseline unbiased AL models, our model can achieve better or similar results under various task scenarios.

Our model perform particularly well on *R15* and *D31*, while **SWAL-cora** and **SWAL-corb** have 10% performance drop at the end of the accuracy curves on *R15*. Different from **SWAL-prop** that determines the sampling probabilistic scheme by label uncertainty alone, **SWAL-cora&corb** compute the sampling probabilities by the location of data points in the feature space and account for additional information captured by the Hessian of the total loss and the gradients of the individual losses and predictions. However, on *R15*, the location information even hinders the judgement since there are some clusters that are close to each other on *R15* and hard to be classified. Similar trends are observed in *D31*. There are no significant differences between our model and baselines unbiased AL models on *Tic-tac-toe* – we think this is because the result (0.76~0.77) is already close to the optimal performance that AL can achieve. Since this dataset is fairly imbalanced, AL needs more data to determine the actual decision boundary and overstep the local optimum.

In summary, our approach provides competitive experimental results for various data topologies, and our proposed method effectively improves the basic AL models' performance, achieving more stable and faster convergence rate than the baseline unbiased/de-biased AL methods.

## 5 CONCLUSION

In this paper, we discuss how the bias problems (i.e., *sampling bias*, *dataset shift*) arise from the sample selection and model fitting steps during the AL processes. We then explore crucial statistical properties (i.e., asymptotically unbiasedness, asymptotically efficiency and consistency) for designing AL approaches that reduce the negative effects of bias problems. Based on these considerations, we propose a flexible AL framework that operates on the top of

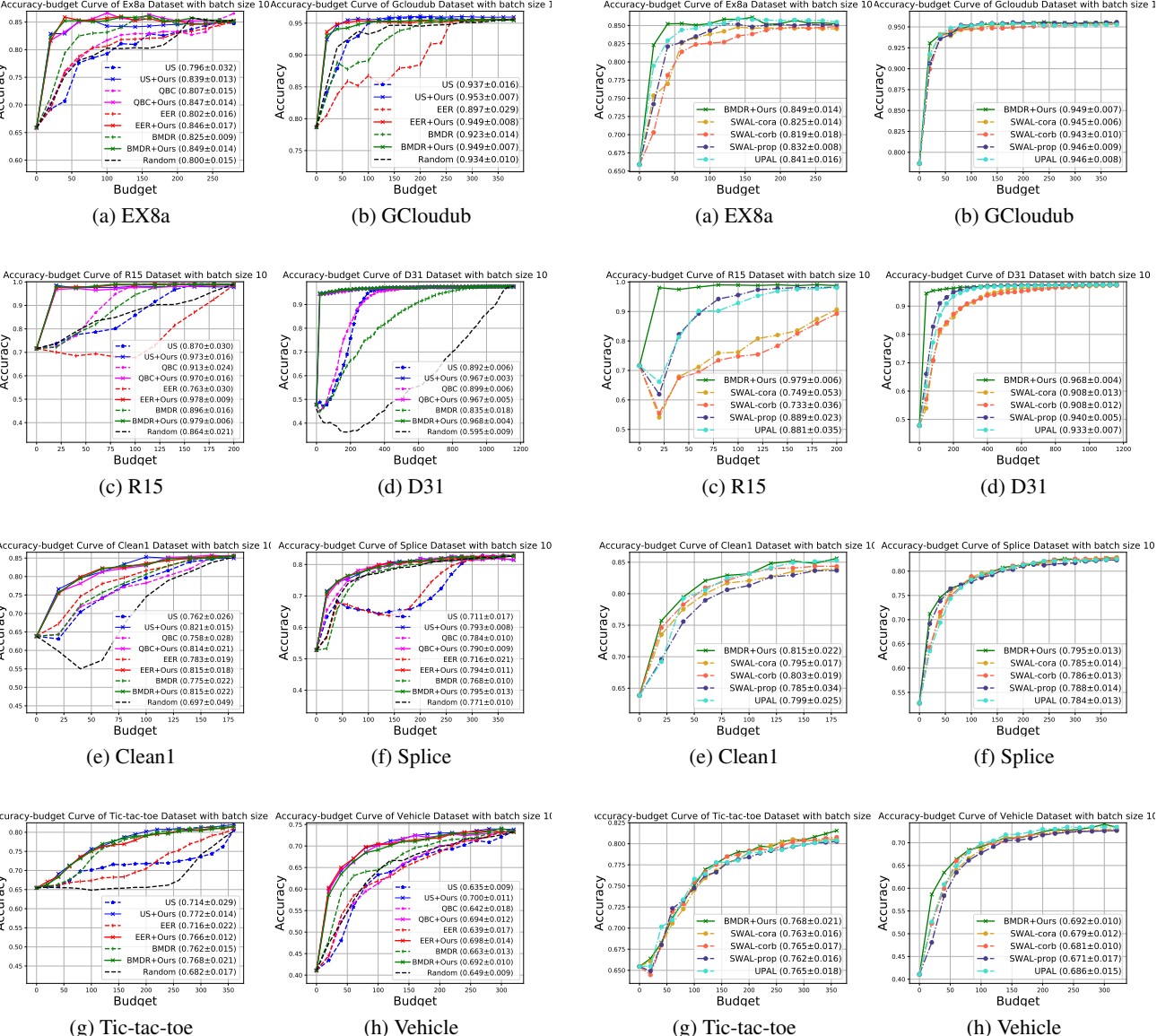

Figure 1: Accuracy-budget curves for classical ML tasks with $B = 10$, including the comparison between our framework with basic AL methods (i.e., **US**, **QBC**, **EER** and **BMDR**). The solid lines represent our methods and dashed lines represent the corresponding baseline AL methods.

Figure 2: Accuracy-budget curves for classical ML tasks with $B = 10$, including the comparison between our framework (we select **BMDR** as basic AL for comparison) and unbiased AL baselines, i.e., **SWAL**, **UPAL**.

existing AL sampling schemes. It provides asymptotically unbiased, efficient and consistent estimate of true risk by utilizing *sampling bias* and well modeling *dataset shift*. The experimental results show that the proposed framework improves the generalization of various basic AL models and also maintains a certain advantage on various data topologies, comparing with other unbiased/de-biased AL sampling schemes.

## Acknowledgements

This work was supported by a grant from the Research Grants Council of the Hong Kong Special Administrative Region, China (Project No. CityU 11215820).

The authors would like to thank dear friends Dr. Xinhong Chen and Dr. Hui Lan from City University of Hong Kong for their useful discussions and feedback.

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
