# OpenReview forum: "Asymptotic Optimality for Active Learning Processes"
_auai.org/UAI/2022/Conference — UAI 2022 Poster_

### Official Review · Reviewer_YZmS · 2022-04-09

**Q2(1) Originality/Novelty:** 2
**Q2(2) Significance/Impact:** 2
**Q2(3) Correctness/Technical Quality:** 2
**Q2(6) Clarity Of Writing:** 2
**Q6 Overall Score:** 5
**Q8 Confidence In Your Score:** 3

**Q1 Summary And Contributions:**

This paper first proposes an AL framework with asymptotic unbiasedness, with respect to basic estimators when the sample size becomes large, and in the limit as the sample size tends to infinity. According to experiments on classic ML tasks, the proposed AL framework has a higher prediction performance and convergence rate.

**Q2 Assessment Of The Paper:**

More detailed information regarding each of these aspects is given below:

**Q2(4) Quality Of Experiments (Optional):**

3: Good: The experimental evaluation is adequate, and the results convincingly support the main claims.

**Q2(5) Reproducibility:**

3: Good: Key resources (e.g., proofs, code, data) are available and key details (e.g., proofs, experimental setup) are sufficiently well-described for competent researchers to confidently reproduce the main results.

**Q3 Main Strengths:**

1.	Theoretic analysis is conducted to show the convergence of the proposed AL framework when the number of iterations approaches infinity.
2.	The proposed AL framework is extendable because it doesn't require predefined sampling and dataset distributions (Q and P).


**Q4 Main Weakness:**

1.	According to the theoretical analysis, the unbiased estimation doesn’t establish until the number of sampling approaches infinity. However, when the number of sampling approaches infinity, what’s the advantage of the AL framework compared with classic ML algorithms.
2.	It seems that in this framework, all sampled data should be saved and used in the calculation of P, Q, and a_t. Besides, this paper didn’t compare the proposed method’s computation complexity with other AL methods.


**Q5 Detailed Comments To The Authors:**

1. The second to last paragraph in section 3.3 is confusing.

**Q7 Justification For Your Score:**

To estimate probabilities in the proposed AL framework, a large space is needed to save sampled samples, and heavy calculations are needed to fit trainable parameters. Authors should put algoirhtms' running time into the paper.

**Q9 Complying With Reviewing Instructions:**

1: Yes.

---

### Official Review · Reviewer_YKx9 · 2022-04-09

**Q2(1) Originality/Novelty:** 2
**Q2(2) Significance/Impact:** 2
**Q2(3) Correctness/Technical Quality:** 1
**Q2(6) Clarity Of Writing:** 1
**Q6 Overall Score:** 3
**Q8 Confidence In Your Score:** 4

**Q1 Summary And Contributions:**

see Q5

**Q2 Assessment Of The Paper:**

More detailed information regarding each of these aspects is given below:

**Q2(4) Quality Of Experiments (Optional):**

2: Fair: The experimental evaluation is weak: important baselines are missing, or the results do not adequately support the main claims.

**Q2(5) Reproducibility:**

2: Fair: Key resources (e.g., proofs, code, data) are unavailable but key details (e.g., proof sketches, experimental setup) are sufficiently well-described for an expert to confidently reproduce the main results.

**Q3 Main Strengths:**

see Q5

**Q4 Main Weakness:**

see Q5

**Q5 Detailed Comments To The Authors:**

The authors claim to derive an active learning framework that is asymptotically unbiased. The key idea is to introduce an instrumental distribution Q that represents the intent of the query strategy. Based on the distribution, the authors take the technique of re-weighting to correct the discrepancy between Q and P (the true distribution of interest). The authors discuss how one can convert existing active learning strategies to an (approximate) instrumental distribution Q_t in every iteration. The conversion and the assumption of non-informativeness lead to a claimed unbiased process asymptotically. Experimental results demonstrate that the discrepancy correction can improve existing AL strategies.

The reviewer reviewed an earlier submission of this paper for another conference. The authors have attempted to address many of the issues from the previous submission, which should be appraised. Admittedly, several of the design thoughts, such as using an approximate instrumental distribution Q_t in every iteration, are certainly interesting and could be valuable for the community.

Yet, the paper still suffers from the following issues:

* The paper is based on the assumption that an instrumental distribution Q exists. After reading the author's update in footnote 1 (page 3), it is still not clear to me why Q must exist. The authors only hand-wavingly say that they consider "meaningful" Q, but even for a reasonable AL process, such as a typical clustering-based uncertainty sampling, the probability that each cluster is sampled depends on how much the cluster has been sampled in the past. In this sense, the sampling probability for each cluster would keep fluctuating (bigger if the cluster has been sampled less, and than smaller if the cluster has been sampled more). The fluctuating nature does not physically guarantee convergence to Q nor the existence of Q. The authors at least need to rigorously define what they mean by "reasonable" AL or "meaningful" Q, with math, not adjectives.

* Following some previous issues, the authors now also assume AL will not query non-existent or out-of-distribution (OOD) data samples. I assume that it means AL will only query when P(x, y) > 0 but this can be spelled out. The authors also assume that "oracles/experts will not produce wrong/noisy labels", but this is an important and *very restricted* assumption as it effectively assumes a clean distribution with P(y | x) = f*(x) for some clean function f*. The assumption is a serious limitation of the proposed framework (e.g. most likely all data sets used in the experiments do not satisfy this assumption), and should be spelled out and discussed more carefully.

* The paper is far from rigorous in theory. For instance, after Assumption 1, the authors discuss the convergence of the q function when the number of queries go to infinity, when |D_u| (the number of unlabeled data) is clearly finite. It is not clear what it means to have a finite-sized labeled pool while pushing the number of queries to infinity.

* On P5, the authors say "in the ideal case, as sample size tends to infinity, enough data is observed and thus the underlying data distribution P is known". This is not a rigorous claim. Perhaps a more rigorous claim is P_t can approximate P well (if the Bayesian approximation is good enough---that's a big assumption as well). Then the authors move on to say "the optimal sampling distribution should be the data distribution itself". This is again not a rigorous claim nor obviously correct. Even if we have a super big data set sampled from P, it is not clear whether we should sample another example from P directly for "optimal" performance (actually, it is not clear what the authors mean by optimal here). This needs to be further clarified.

* I read the authors' discussions in Appendix 3.2 for the cases of non-informativeness. The authors honestly state that some acquisition functions cannot achieve non-informativeness, which should be appraised. The issue is arguably an important aspect of this work and deserves to be discussed in the main paper rather than the supplementary material. In addition, the fact that the authors' claimed non-informativeness can only be achieved by entropy sampling *on clean data* instead of entropy sampling in general should be seriously highlighted.

* Overall, the authors seem to be adding assumptions in a non-rigorous and overly-flexible manner (e.g. Assumption 1 is formal, but things like "AL will not query non-existent or out-of-distribution (OOD) data samples" is ad-hoc). This, along with many claims that are not well-supported (e.g. "the optimal sampling distribution should be the data distribution itself"), makes the theoretical validity and rigor questionable. The authors are suggested to improve the writing with higher quality logical reasoning: what are being assumed? are the assumptions physically reasonable? what do the assumptions imply? The mathematical quality of the current paper still cannot be trusted.

* On a bird's eye view, it seems that the authors assume that if we can get infinite number of samples, and AL's Q_t approximates a Q that is non-informative (constant) as we get more and more samples. Then the empirical risk of the infinite number of samples is unbiased. My question is, is this the intended reasoning? If so, is it trivial? I mean, this essentially says "If AL is assumed to approximate uniform sampling when getting infinite number of examples, the empirical risk, like uniform sampling, is unbiased." While the statement is true, there is very little technical difficulty (after placing a very strong assumption on non-informativeness). Can the authors explain why they believe the results to be non-trivial?


**Q7 Justification For Your Score:**

see Q5---mainly, as a theory paper, the math quality still cannot be trusted.

**Q9 Complying With Reviewing Instructions:**

1: Yes.

---

### Official Review · Reviewer_TPax · 2022-04-14

**Q2(1) Originality/Novelty:** 3
**Q2(2) Significance/Impact:** 3
**Q2(3) Correctness/Technical Quality:** 3
**Q2(6) Clarity Of Writing:** 3
**Q6 Overall Score:** 6
**Q8 Confidence In Your Score:** 4

**Q1 Summary And Contributions:**

In this paper, the authors have the theoretical properties of active learning. They have shown the crucial factors that can help active learning reduce the negative effects of sampling bias. An active learning framework has also been proposed to reduce sampling bias.

**Q2 Assessment Of The Paper:**

More detailed information regarding each of these aspects is given below:

**Q2(4) Quality Of Experiments (Optional):**

3: Good: The experimental evaluation is adequate, and the results convincingly support the main claims.

**Q2(5) Reproducibility:**

3: Good: Key resources (e.g., proofs, code, data) are available and key details (e.g., proofs, experimental setup) are sufficiently well-described for competent researchers to confidently reproduce the main results.

**Q3 Main Strengths:**

1.	The authors have provided a perspective on AL in terms of statistical properties. Specifically, the authors have shown empirical risk minimization is biased in active learning. To solve the bias issues important reweighting estimation can be employed.
2.	A method is proposed to which are conceptually better than existing methods.
3.	The paper is easy to follow and is ordered in a logical way.


**Q4 Main Weakness:**

1. Variables $\beta$, $\alpha$, and $c$ could be hard to accurately estimate without any assumptions, and the assumptions may not be fulfilled in practice.

**Q5 Detailed Comments To The Authors:**

This paper looks novel to me. I hope some of my suggestions can help to improve this paper, which are as follows.

1. The importance weighting may not work in some cases, and some conditions may be required to be satisfied. Because the method proposed is based on the importance of reweighting. It would be great to show clearly illustrate the limitation of the importance of weighting.

2. Empirically, $P_t$ is used to approximate $P$. It would be great to add some discussion on the feasibility and influence of such estimation.

3. In Table 1, it is would be great if the authors can also illustrate the standard deviation of accuracies.


**Q7 Justification For Your Score:**

I weigh 1 and 2 in Main strengths most heavily. I think the takeaway message delivered by this paper can be useful.
The drawback is that a lot of variables can be hard to be estimated if the assumptions do not satisfied .

**Q9 Complying With Reviewing Instructions:**

1: Yes.

---

### Official Review · Reviewer_d2Km · 2022-04-16

**Q2(1) Originality/Novelty:** 3
**Q2(2) Significance/Impact:** 2
**Q2(3) Correctness/Technical Quality:** 3
**Q2(6) Clarity Of Writing:** 3
**Q6 Overall Score:** 6
**Q8 Confidence In Your Score:** 2

**Q1 Summary And Contributions:**


This paper proposes an Active Learning framework to mitigate the overfitting during active learning such as sampling bias and dataset shift. They propose a framework that enjoys some asympototic unbiasedness as the sample size increases.
Additionally, in few sets of experiments they showed advantage of their method in out-performing few other Active Learning baselines.


**Q2 Assessment Of The Paper:**

More detailed information regarding each of these aspects is given below:

**Q2(4) Quality Of Experiments (Optional):**

3: Good: The experimental evaluation is adequate, and the results convincingly support the main claims.

**Q2(5) Reproducibility:**

3: Good: Key resources (e.g., proofs, code, data) are available and key details (e.g., proofs, experimental setup) are sufficiently well-described for competent researchers to confidently reproduce the main results.

**Q3 Main Strengths:**

- Overall the paper is well written, and it was easy to follow for me.
- Related work section touches upon many similar works and does a decent job motivating and explaning the differences with their method

- The theoretical advantages of the proposed framework such as asympototic unbiasedness seem interesting, and rigourous proofs are provided, did not read the proofs in too much detail but from a quick glance they look correct.



**Q4 Main Weakness:**

- Figure 1, there is a lot going on in the figures, and fonts are very small. It's very hard to see what's happening even after zooming in.
I suggest increasing the fonts and simplifying the figures a bit. Maybe seperate the figures and be more selective with what's included in the main paper.

- In few places, more technical details would have helped. There is a lot hyperparamters or customiazations possiblei in the framework.
For example, what are the limitations on how to choose Q_t ,P_t? Equation 11,13 involve taking potentially intractable integrals? Do you only choose q, P in a way that the integral is tractable? Or do you approximate the integrals? In that case, how does the approximations play into each other for the overall framework

- Not necessarily a big issues, but having multiple thresholds for p-value in table 1 felt really odd to me.

**Q5 Detailed Comments To The Authors:**

- In theorem 1, is it guaranteed a, b are finite?

- What model is used to estimate p_t, q_t, Q_t, P_t, etc. Is there parameter sharing between the `p_t` for different t's, or each model is indpedendent?

- The convergence to "non-informative" aquition function was interesting, what is the main advantage of being non-informative? Is it to correct for sampling bias or the dataset shift bias?

- In practice, how easy is it to come up with a strategy that "Assumption 1" holds??


**Q7 Justification For Your Score:**

Overall there was good things I liked about the paper. That being said, there were few aspects that could see improvements to make things more clear, such as better presentation for figure 1.

**Q9 Complying With Reviewing Instructions:**

1: Yes.

---

### Decision · Program_Chairs · 2022-05-15

**Decision:**

Accept (Poster)

**Comment:**

Meta Review: I find this paper to present an interesting perspective, analyzing the sampling bias in active learning and correcting for it.  There was some debate with Review YKx9 about the assumptions and rigor of the theory, but I found the author responses adequate.  I recommend acceptance.  The authors should take care to improve the readability of Figure 1.